# Sentiment-oriented Sarcasm Integration for Video Sentiment Analysis Enhancement with Sarcasm Assistance

Junlin Fang
Southwest Jiaotong University
School of Computing and Artificial Intelligence
Chengdu, China
jlfang@my.swjtu.edu.cn

Wenya Wang
Nanyang Technological University
School of Computer Science and Engineering
Singapore, Singapore
wangwy@ntu.edu.sg

Guosheng Lin
Nanyang Technological University
School of Computer Science and Engineering
Singapore, Singapore
gslin@ntu.edu.sg

Fengmao Lv*
Southwest Jiaotong University
School of Computing and Artificial Intelligence
Chengdu, China
fengmaolv@swjtu.edu.cn

## Abstract

Sarcasm is an intricate expression phenomenon and has garnered increasing attentions over the recent years, especially for multimodal contexts such as videos. Nevertheless, despite being a significant aspect of human sentiment, the effect of sarcasm is consistently overlooked in sentiment analysis. Videos with sarcasm often convey sentiments that diverge or even contradict their explicit messages. Prior works mainly concentrate on simply modeling sarcasm and sentiment features by utilizing the Multi-Task Learning (MTL) framework, which we found introduces detrimental interplays between the sarcasm detection task and sentiment analysis task. Therefore, this study explores the effective enhancement of video sentiment analysis through the incorporation of sarcasm information. To this end, we propose the Progressively Sentiment-oriented Sarcasm Refinement and Integration (PS2RI) framework, which focuses on modeling sentiment-oriented sarcasm features to enhance sentiment prediction. Instead of naively combining sarcasm detection and sentiment prediction under an MTL framework, PS2RI iteratively performs the sentiment-oriented sarcasm refinement and sarcasm integration operations within the sentiment recognition framework, in order to progressively learn sarcasm-aware sentiment feature without suffering the detrimental interplays caused by information irrelevant to the sentiment analysis task. Extensive experiments are conducted to validate the effectiveness of our approach. Code is available at https://github.com/tiggers23/PS2RI.

## CCS Concepts

• **Information systems** → **Sentiment analysis**; **Multimedia information systems**.

*Corresponding author.

## Keywords

Video sentiment analysis; sarcasm detection; feature fusion; multimodal deep learning.

**ACM Reference Format:**
Junlin Fang, Wenya Wang, Guosheng Lin, and Fengmao Lv. 2024. Sentiment-oriented Sarcasm Integration for Video Sentiment Analysis Enhancement with Sarcasm Assistance. In *Proceedings of the 32nd ACM International Conference on Multimedia (MM '24), October 28-November 1, 2024, Melbourne, VIC, Australia.* ACM, New York, NY, USA, 10 pages. https://doi.org/10.1145/3664647.3680703

## 1 Introduction

Sarcasm detection has gained significant attentions in recent years due to the potential impact of sarcasm on the sentiment analysis task [20, 21, 25, 32, 38, 55]. The key factor of sarcasm detection is to extract incongruous features which can suggest the distinction between the conveyed sentiment and the author's intended sentiment [25, 32, 55]. The contradictory sentiments arising from such incongruity can considerably disrupt the sentiment analysis task. Recent research [15] has consistently shown that highly advanced Large Language Models (such as ChatGPT [31]), face challenges when it comes to detecting genuine sentiments conveyed in sarcastic samples. Therefore, it is crucial for sentiment analysis systems to handle sarcasm which is usually overlooked by existing works.

In practice, sarcasm can become more prevalent within multimodal inputs like videos [12, 24, 29, 39, 42, 56], which convey contradicting sentiments among different modalities. As shown in Figure 1, the text conveys a positive sentiment, while the video demonstrates a negative sentiment which reflects the intrinsic sentiment of the character. Existing studies implement multimodal sarcasm detection by modeling the incongruity information within different aspects of multimodal contents [4, 21, 25, 32]. However, there is a lack of investigation on how sarcasm information can be effectively utilized to enhance the sentiment analysis task.

To this end, an intuitive way is to utilize the Multi-Task Learning (MTL) framework by jointly training the sarcasm detection and sentiment analysis tasks, which has been studied by Chauhan et al. [4] and Liu et al. [26]. The MTL framework is expected to enhance sentiment prediction by learning shared representations across the two tasks. However, as will be shown in Section 5, after a

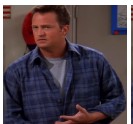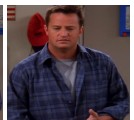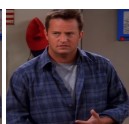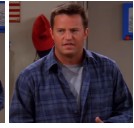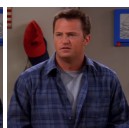

Well thank God your livelihood doesn't depend on it.

**Figure 1: An example of video with sarcasm. The text of this video segment is positive, while the facial sentiment expressed in the visual modality is negative.**

comprehensive evaluation, it turns out that the MTL framework impedes sentiment prediction over non-sarcastic inputs which can be easily identified using a single sentiment classifier (see Table 1). This phenomenon implies that the sentiment analysis task suffers the negative interference of the sarcasm detection task in the MTL framework, which is a universally present challenge in MTL [13, 52].

Motivated by the above observation, we propose the Progressive Sentiment-oriented Sarcasm Refinement and Integration (PS2RI) mechanism which boosts video sentiment prediction capability with sarcasm-aware information. Figure 2 shows the model overview of the proposed PS2RI approach. The core idea of PS2RI is to iteratively perform the sentiment-oriented sarcasm refinement and sarcasm integration operations within the sentiment recognition framework, in order to discard irrelevant signals of sarcasm features during the sarcasm-aware sentiment learning process. To this end, our model first utilizes two separate encoders to respectively extract sarcasm-informed features and sentiment-related features from videos. Then a Sarcasm-Aware Sentiment Learning (SASL) module is stacked on top of sentiment-related features, with the purpose of producing sarcasm-aware sentiment representations. SASL is implemented via an iterative framework in order to progressively explore the deep interactions between two tasks. Specifically, SASL iteratively propagates information between the two tasks by first refining sarcasm features using sentiment information, and then fusing the refined sarcasm features obtained in the previous step into the sentiment feature learning process. Within each iteration, the sarcasm refinement operation is introduced to obtain sentiment-oriented sarcasm features, to reduce the negative interference caused by direct incorporation of sarcasm features. Compared to the MTL framework [4] which shares features across tasks, PS2RI can alleviate the negative interference of sarcasm detection by discarding irrelevant signals of sarcasm features via sentiment-oriented sarcasm learning before fusing them into the sentiment learning module, thus attentively propagating effective signals between sarcasm and sentiment throughout the learning process (as shown in the supplementary). The superiority of PS2RI is verified via extensive experiments on different video sentiment analysis benchmarks.

To sum up, the contributions of this work can be summarized as follows:

- We conduct a comprehensive study on how to improve sentiment analysis with the assistance of sarcasm detection and discuss the drawbacks of existing MTL-based methods.
- We propose the PS2RI framework which effectively alleviates the negative interference caused by the interaction of sentiment analysis and sarcasm detection; and enhances the performance of sentiment prediction significantly.
- We conduct extensive evaluations to demonstrate the effectiveness and scalability of our proposed PS2RI framework.

## 2 Relate Work

### 2.1 Video Sentiment Analysis

Video sentiment analysis is primarily focused on identifying the sentiment polarity of humans from videos. Given the presence of different information across various modalities, such as language, acoustic, and visual, video understanding requires fusing data from different modalities [6, 10, 19, 30, 43, 45, 60, 64]. In general, existing video sentiment analysis methods can be divided into global-level fusion and element-level fusion. Specifically, the former strategy implements multimodal fusion over global-level feature vectors extracted from each individual modality and does not consider the cross-modal interactions between elements [17, 35, 42, 43, 48, 50, 51]. Compared to global-level fusion, the element-level fusion methods can achieve a more thorough inter-modal interaction by modeling the fine-grained feature fusion with word-level information considered [22, 23, 28, 45, 49, 65]. However, the above works mainly focus on multimodal fusion, ignoring a further investigation from the sentiment recognition aspect. As indicated by recent works [21, 25], sarcasm information usually makes a negative interference on the sentiment analysis task.

### 2.2 Multimodal Sarcasm Detection

Besides the standard sentiment recognition task as introduced above, there also exit various sentiment-related recognition problems, such as metaphor detection [53, 61, 62], humor recognition [5, 11, 33, 54], and sarcasm detection [12, 24, 29, 39, 42, 56]. In this work, we mainly focus on the multimodal sarcasm detection problem. Sarcasm detection aims to capture the contradictory sentiment information present in samples, which has garnered significant attentions due to its effect on causing a shift or complete reversal in the expressed sentiment, thereby greatly impacting sentiment analysis [32, 38, 55]. Early sarcasm detection works primarily focuses on text-only samples, with the core idea being to identify incongruous patterns within texts [9, 37, 41]. Recently, there has been a surge of interest in multimodal sarcasm detection, driven by the growing fascination people have with posting and browsing multimodal information [40]. Compared to text-based sarcasm detection, multimodal sarcasm detection can be more difficult to identify considering the misalignment across different modalities. Existing works on multimodal sarcasm detection focus on exploring incongruity information within different aspects of multimodal contents by building graph neural networks [21, 25] or modeling cross-modal discrepancies and intra-text incongruities [32]. Recently, multimodal sarcasm detection has also been explored for videos [2, 63].

However, the above works study the sarcasm detection task independently and there are currently limited studies that explore how sarcasm information enhances the sentiment analysis task, which holds greater importance in practical application scenarios. There are some studies that acknowledge the interaction between sentiment analysis and sarcasm detection, employing a MTL framework to jointly train the sarcasm detection and sentiment analysis tasks, leading to good advances in sentiment analysis [4, 26]. Nevertheless, their primary emphasis resides in amalgamating features from these two tasks, lacking a more comprehensive evaluation for

the sentiment analysis task. Moreover, by conducting a comprehensive evaluation, we find that the MTL framework can only make a marginal improvement for sentiment analysis, and even hurts the performance on non-sarcastic examples.

## 3 Problem Definition & Preliminary

### 3.1 Problem Definition

The task of video sentiment analysis involves three different modalities, i.e., language (l), vision (v), and acoustic (a) modalities. Following Chauhan et al. [4], we denote a sample in training data as $X = \{(U, C), Y_{sar}, Y_{sen}\}$, where $U$ and $C$ correspond to the target utterance and associated historical contextual dialogues, respectively. $Y_{sar}$ denotes the binary sarcasm label and $Y_{sen}$ denotes the sentiment label. To be specific, we use $U_m \in \mathbb{R}^{T_m^u \times d_m}$, where $m \in \{l, v, a\}$ refers to a specific modality, to denote the utterance feature from modality $m$. $T_m^u$ and $d_m$ denote the utterance length and feature dimension, respectively. Similarly, the context feature can be denoted by $C_m \in \mathbb{R}^{T_m^c \times d_m}$, where $T_m^c$ denotes the context length. The core idea of this problem is to integrate data of the three modalities and learn sarcasm-aware sentiment representations which are reinforced by sarcasm information.

### 3.2 Cross-modal Attentions

The cross-modal attention operation fuses the source modality information into the target modality [46]. We denote $X_s \in \mathbb{R}^{T_s \times d_s}$ the source modality feature and $X_t \in \mathbb{R}^{T_t \times d_t}$ the target modality feature. The cross-modal attention mechanism treats the source modality as keys and values to be attended using the target modality as the query:

$$\begin{aligned} Y_{s \to t} &= \mathrm{CA}(X_s, X_t) \\ &= \mathrm{softmax}(\frac{(X_t W_q) \cdot (W_k^\top X_s^\top)}{\sqrt{d_k}}) X_s W_v, \end{aligned} \quad (1)$$

where $W_q, W_k$ and $W_v$ are transformation matrices. $Y_{s \to t} \in \mathbb{R}^{T_t \times d_t}$ is the resulting representation in the target domain. For ease of illustration, we use CA to denote the cross-modal attention process as shown in Eq. (1) with multiple attention heads. Note that CA can also be used for fusing information from two source modalities to the target modality.

### 3.3 Modality-guided Trimodal Fusion

The Modality-guided Trimodal Fusion (MTF) strategy is introduced in [23] which aims to integrate information from other modalities into each unimodal feature representation. Specifically, MTF consists of multiple layers, and we denote by $\mathrm{MTF}_m^n$ the $n$-th layer for modality $m \in \{l, v, a\}$. Given input features $Z_l^{n-1}, Z_v^{n-1}, Z_a^{n-1}$ from three modalities at the $(n-1)$th layer, we denote by

$$Z_m^n = \mathrm{MTF}_m^n(Z_l^{n-1}, Z_v^{n-1}, Z_a^{n-1}) \quad (2)$$

the output feature for modality $m$ after trimodal fusion. More specifically, $\mathrm{MTF}_m^n$ is computed by firstly applying CA between the target modality (e.g., $m = l$) and the other two modalities using Eq. (1). Then a gate unit is applied to combine the faetures obtained from two cross-modal attention operations i.e., $Y_{v \to l} = \mathrm{CA}(Z_v^{n-1}, Z_l^{n-1})$ and $Y_{a \to l} = \mathrm{CA}(Z_a^{n-1}, Z_l^{n-1})$, as proposed in [23].

In order to combine features from three modalities as the final representation, a gate [28] is further applied on top of the last MTF layer by weighing the contribution from each modality $m$:

$$\begin{aligned} \mu_m &= U^\top \tanh(Z_m^N \cdot W_m + b_m), \\ \alpha_m &= \frac{\exp(\mu_m)}{\sum_{m' \in \{l,v,a\}} \exp(\mu_{m'})}, \\ Z &= \sum_{m \in \{l,v,a\}} \alpha_m \odot Z_m^N, \end{aligned} \quad (3)$$

where $Z_m^N$ is the output of the last MTF layer i.e., $\mathrm{MTF}_m^N$. $U^\top$ are learnable parameters, $W_m$ and $b_m$ are the parameters of a linear function. For ease of illustration, we use $Z = \mathcal{F}(Z_l^0, Z_v^0, Z_a^0)$ to refer to the fusion output given by Eq. (2) and (3).

## 4 Methodology

### 4.1 Model Overview

Figure 2 shows the overall architecture of our proposed PS2RI framework. Given a video input consisting of textual, acoustic and visual features, the PS2RI model first utilizes two separate encoders (i.e., Sarcasm Feature Encoder and Sentiment Feature Encoder) to respectively produce sarcasm-informed features and sentiment-related features. A Sarcasm-Aware Sentiment Learning module is then stacked on top of sentiment-related features, with the purpose of producing sarcasm-aware sentiment representations for enhanced sentiment recognition. SASL is implemented via an iterative framework to progressively explore the deep interactions between two tasks. Within each iteration, SASL utilizes a Sentiment-Oriented Sarcasm Refinement (SOSR) operation to refine sarcasm features guided by relevant sentiment information, which is then followed by a Sarcasm Integration (SI) operation to merge the sentiment-oriented sarcasm feature into the sentiment feature learning process. After implementing the above progressive refinement and integration process, SASL will result in sarcasm-aware sentiment features, which can lead to enhanced sentiment recognition performance. The PS2RI framework provides an effective way to alleviate the negative interference caused by the detrimental interplay between the sentiment analysis and sarcasm detection tasks (as shown in the supplementary).

### 4.2 Feature Preprocessing

In this work, we adopt the same feature extraction step used in [11] to extract features from each modality. To extract acoustic features, we preprocess acoustic frames using COVAREP [7] which is capable of processing Melcepstral coefficients, fundamental frequency, voiced/unvoiced segments, normalized amplitude quotient, quasi open quotient [14], glottal source parameters [8], harmonic model, phase distortions, and the formants. For the visual modality, facial Action Unit (AU) features and rigid/non-rigid facial shape parameters are extracted using OpenFace 2 [1]. For the textual modality, we use the ALBERT tokenizer [16] to convert language utterances, following the previous work [11]. Formally speaking, we denote by $X_l = [C_l; U_l]$, $X_v = [C_v; U_v]$ and $X_a = [C_a; U_a]$ input feature for the language, vision and acoustic modality respectively, where $C$ and $U$ indicate context and utterance respectively. We denote $X_m \in \mathbb{R}^{(T_m^u + T_m^c) \times d_m}$ one of the three modalities, where $m \in \{l, v, a\}$.

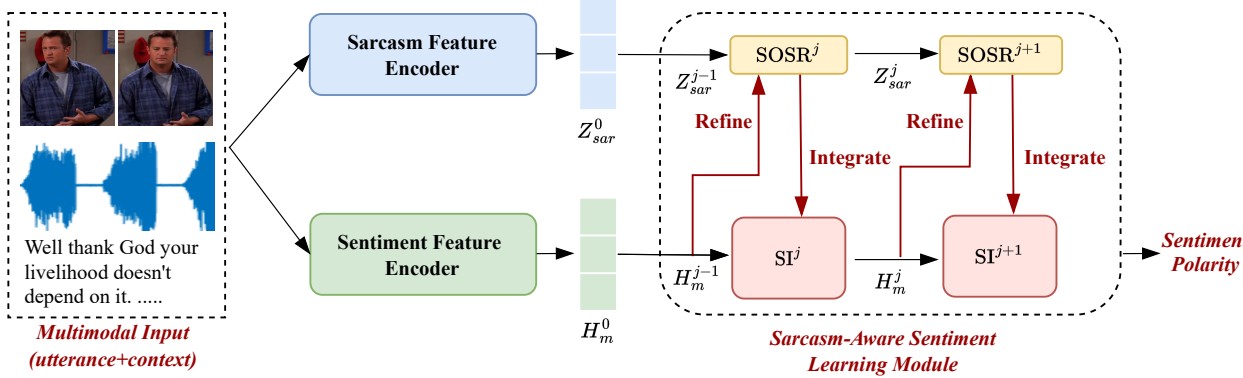

**Figure 2: The model overview of the proposed PS2RI framework. The Sarcasm Feature Encoder and Sentiment Feature Encoder will first produce sarcasm-informed features $Z_{sar}^0$ and sentiment-related features $H_m^0$, respectively. The Sarcasm-Aware Sentiment Learning module is stacked on top of $H_m^0$ to produce sarcasm-aware sentiment representations by iteratively integrating information from the sarcasm-informed feature. Within each iteration of the SASL, the sarcasm feature $Z_{sar}^{j-1}$ will first be refined by the sentiment features $H_m^{j-1}$ in SOSR$^j$, and then be integrated into the sentiment feature $H_m^j$ via SI$^j$, resulting in $H_m^j$. By iteratively performing the refinement and integration stages, the SASL module will finally produce sarcasm-aware sentiment feature $H_m^J$ for enhanced sentiment recognition.**

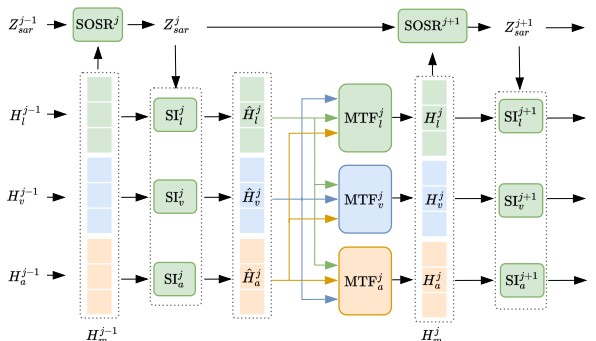

**Figure 3: The detail of each SASL iteration. Within each iteration, the Sentiment-Oriented Sarcasm Refinement (SOSR), Sarcasm Integration (SI) and Modality-guided Trimodal Fusion (MTF) operations are involved.**

### 4.3 Sarcasm/Sentiment Feature Encoding

**Sarcasm Feature Encoder.** The Sarcasm Feature Encoder aims to discriminate sarcasm-related information from inputs without any interference from the sentiment aspect. To achieve that, we first process the multimodal input using their respective encoders. Following [11], for textual input $X_l$, we use a pre-trained ALBERT encoder composed of 12 layers, together with a linear transformation layer, to produce the text representation $H_l^{sar} = \text{Linear}_l^{sar}(\text{Albert}^{sar}(X_l))$, where $H_l \in \mathbb{R}^{(T_m^u + T_m^c) \times d^H}$ and $d^H$ is the output dimension of the alignment layer which maps different modalities features into the same space. Following [11], we apply a multi-layer transformer encoder [47] and a linear alignment layer to process the visual modality $H_v^{sar} = \text{linear}_v^{sar}(\text{Transformer}_v^{sar}(X_v))$ and acoustic modality $H_a^{sar} = \text{linear}_a^{sar}(\text{Transformer}_a^{sar}(X_a))$, where $H_v^{sar}, H_a^{sar} \in \mathbb{R}^{(T_m^u + T_m^c) \times d^H}$. Note that we use the superscript "sar" to denote the functions and variables within the sarcasm feature extraction module.

Given transformed features $H_l^{sar}, H_v^{sar}$ and $H_a^{sar}$ from three modalities, we adopt the Modality-guided Trimodal Fusion (MTF)

strategy introduced in Section 3.3 to generate a unified cross-modal feature representation $Z_{sar} = \mathcal{F}(H_l^{sar}, H_v^{sar}, H_a^{sar})$ via Eq. (2) and (3). We train this module with the supervision of the sarcasm label $Y_{sar}$ by minimizing the cross-entropy loss between $Y_{sar}$ and the predicted score $Y'_{sar} = f(Z_{sar})$, where $f$ is composed of a 1D max pooling layer and a linear layer. As a result, this sarcasm feature encoder can produce sarcasm-discriminative features.

**Sentiment Feature Encoder.** Similar to the Sarcasm Feature Encoding module, we use another set of parameters to process the multimodal inputs, and denote them using the superscript "sen" which indicates sentiment: $H_l^{sen} = \text{Linear}_l^{sen}(\text{Albert}^{sen}(X_l))$, $H_v^{sen} = \text{linear}_v^{sen}(\text{Transformer}_v^{sen}(X_v))$, $H_a^{sen} = \text{linear}_a^{sen}(\text{Transformer}_a^{sen}(X_a))$, where $H_l^{sen}, H_v^{sen}$, and $H_a^{sen} \in \mathbb{R}^{(T_m^u + T_m^c) \times d^H}$. Unlike the sarcasm feature encoder, the sentiment feature encoder does not consider the trimodal fusion of the three modalities and leaves the fusion process to the following Sarcasm-Aware Sentiment Learning procedure, due to the computation overhead and model complexity aspects. We treat the above generated features $H_{\{l,v,a\}}^{sen}$ as sentiment-related features, which will then be input into the SASL module to generate sarcasm-aware sentiment representations.

### 4.4 Sarcasm-Aware Sentiment Learning

With a well-trained sarcasm feature extractor, we stack the SASL module on top of sentiment-related features $Z$ to produce sarcasm-aware sentiment representations by effectively incorporating useful sarcasm features. Within each iteration, two interaction block are involved: a Sentiment-Oriented Sarcasm Refinement (SOSR) block and a Sarcasm Integration (SI) block. Initially, the SOSR block takes the output feature $Z_{sar}$ generated from the Sarcasm Feature Encoding module and provides a refiement given input features coming from the sentiment module. This update infuses original sarcasm-only features with relevant sentiment information which progressively orients sarcasm features towards the sentiment space, reducing the task gap. On the other hand, the SI block receives the

updated sarcasm features and integrates them into the representation of each modality adaptively, thus merging relevant sarcasm features into the sentiment learning process. We alternate these two blocks progressively throughout multiple layers to reinforce the propagation between two task signals. Figure 3 shows the detail process of PS2RI. The algorithm which displays the information flow across the PS2RI procedure, is shown in the supplementary.

**Sentiment-Oriented Sarcasm Refinement.** We use $SOSR^j$ to denote the SOSR block at the $j$-th layer. $SOSR^j$ receives sarcasm features $Z_{sar}^{j-1}$ and unimodal sentiment features $H_m^{j-1}$ ($m \in \{l, v, a\}$) from the previous layer as its input and produces sentiment-oriented sarcasm features $Z_{sar}^j \in \mathbb{R}^{(T_m^u + T_m^c) \times d^H}$:

$$Z_{sar}^j = \text{SOSR}^j(Z_{sar}^{j-1}, H_l^{j-1}, H_v^{j-1}, H_a^{j-1}). \tag{4}$$

At the 1st layer, we initialize unimodal sentiment features as $H_m^0 = H_m^{sen}$ and sarcasm features as $Z_{sar}^0 = Z_{sar}$. Next, we illustrate how $SOSR^j$ is computed. Specifically, we merge unimodal sentiment features $H_m^{j-1}$ into a unified representation $Z_{sen}^{j-1}$ via the gate unit introduced in Eq. (3).

Then a cross-attention layer introduced in Section 3.2 and a self-attention layer introduced in [47] are used to infuse sarcasm features with multimodal sentiment representation $Z_{sen}^{j-1}$ by attending to relevant sentiment features:

$$
\begin{aligned}
\overline{Z}_{sar}^{j-1} &= \text{CA}^j(\text{LN}(Z_{sen}^{j-1}), \text{LN}(Z_{sar}^{j-1})), \\
\hat{Z}_{sar}^{j-1} &= f_{FF}(\overline{Z}_{sar}^{j-1} + Z_{sar}^{j-1}), \\
Z_{sar}^j &= f_{FF}(\text{SA}^j(\text{LN}(\hat{Z}_{sar}^{j-1})) + \hat{Z}_{sar}^{j-1}),
\end{aligned} \tag{5}
$$

where LN, $f_{FF}$ and $SA^j$ represent the layer normalization operation, the feed-forward operation and the $j$-th self-attention layer, respectively. This completes the operation of Eq. (4).

**Sarcasm Integration.** The sentiment-oriented sarcasm feature $Z_{sar}^j$ will in turn be used to reinforce the computation of unimodal sentiment features $H_m^{j-1}$ ($m \in \{l, v, a\}$) via a Sarcasm Integration (SI) block. Formally, the SI block takes $Z_{sar}^j$ and $\{H_l^{j-1}, H_v^{j-1}, H_a^{j-1}\}$ as its input and produces enhanced sarcasm-aware sentiment features $\hat{H}_m^j \in \mathbb{R}^{(T_m^u + T_m^c) \times d^H}$, where $m \in \{l, v, a\}$:

$$\hat{H}_m^j = \text{SI}_m^j(Z_{sar}^j, H_m^{j-1}). \tag{6}$$

Specifically, the function $SI_m^j$ includes a cross-attention layer and a self-attention layer to fuse sentiment-oriented sarcasm features $Z_{sar}^j$ into unimodal sentiment representations $H_m^{j-1}$, similar to Eq. (5). The difference with Eq. (5) is that we replace the input $Z_{sen}^{j-1}$ and $Z_{sar}^j$ with $Z_{sar}^j$ and $H_m^{j-1}$ respectively, and the output is a sarcasm-aware sentiment feature $\hat{H}_m^j$, where $m \in \{l, v, a\}$. As a last step at the $j$-th layer, MTF as shown in Eq. (2) generates updated feature representations for each modality: $H_m^j = \text{MTF}_m^j(\hat{H}_l^j, \hat{H}_v^j, \hat{H}_a^j)$. A gate unit as introduced in Eq. (3) is used to integrate $H_l^J, H_v^J$ and $H_a^J$ into $Z_{sen}$, where $J$ is the index of the last layer. $Z_{sen}$ will be fed into the sentiment classifier for final prediction.

## 5 Experiments

Following the previous works [3, 4], we first demonstrate the superiority of our proposed approach on two standard benchmark datasets (i.e., MUStARD [2] and MUStARD++ [36]) which contain both sarcasm and sentiment labels. For in-depth analysis, we split both MUStARD and MUStARD++ datasets into two groups: sarcastic samples (Subset 1) and non-sarcastic samples (Subset 2).

Considering that the sarcasm label is usually difficult to annotate, we also validate the scalability of the proposed SASL mechanism on wider settings where sarcasm labels are not provided. Specifically, we conduct experiments by employing the SASL mechanism in conjunction with state-of-the-art video sentiment recognition models on CMU-MOSI [59] and CMU-MOSEI [58], which are two extensively used benchmarks in video sentiment analysis. As these two datasets do not have sarcasm labels, we directly use the sarcasm feature extraction module trained on MUStARD++ to generate sarcasm features for their samples.

### 5.1 Dataset

The detailed descriptions for the datasets involved in our experiments are listed as follows.

**MUStARD** [2] comprises 3.68 hours of human conversations and provides 675 video samples (50% of which are sarcastic samples) from popular TV shows such as *Friends*, *The Big Bang Theory*, *The Golden Girls* and *Sarcasmaholics Anonymous*. Its predetermined data partition has 539 samples in the training set, 68 samples in the validation set, and 68 samples in the testing set. Castro et al. [2] manually annotate the sarcastic/non-sarcastic label for each example, while Chauhan et al. [4] re-annotate the MUStARD examples with sentiment labels. Therefore, MUStARD contains both 3-class sentiment labels (i.e., positive, neutral and negative) and 2-class sarcasm labels (i.e., sarcastic and non-sarcastic). Each sample in MUStARD consists of a single sentence with accompanied visual and acoustic segments. In agreement with the prior work [36], we adopt the *Weighted-Average* evaluation metrics, including *Precision*, *Recall* and *F1-score* to evaluate the performance.

**MUStARD++** [36] is a sarcasm dataset consisting of 1,202 samples of video clips (50% of which are sarcastic samples). Its predetermined data partition has 962 samples in the training set, 120 samples in the validation set, and 120 samples in the testing set. Ray et al. [36] extend the samples in MUStARD [2] and re-annotate them with 10-class sentiment labels and 2-class sarcasm labels. As in the MUStARD dataset, each sample in MUStARD++ consists of a single sentence with accompanied visual and acoustic segments. Moreover, each sample also comes with a corresponding context in three modalities. The performance metrics are the same to the ones used in the MUStARD dataset.

**CMU-MOSI** [59] is made by 2,199 samples of short video clips. Following the previous works [6, 48, 51], we split the dataset into 1,284 training samples, 229 validation samples and 686 testing samples. Each sample in the dataset is labeled with a sentiment label ranging from -3 (strongly negative) to 3 (strongly positive). The acoustic and visual sequences are extracted at the receiving frequency of 12.5 and 15 Hz, respectively. As in the previous works [6, 48, 51], the performance is evaluated by the 7-class accuracy (i.e., $Acc_7$), binary accuracy (i.e., $Acc_2$) and F1 score.

**CMU-MOSEI** [58] is a dataset consisting of 22,856 samples of movie review video clips which are collected from the YouTube platform. According to the predetermined data partition of CMU-MOSEI, we

**Table 1: Evaluating the contributions of introducing sarcasm information to sentiment analysis through the MTL framework on MUStARD and MUStARD++, where $\mathcal{F}$-SEN is the model only trained with sentiment supervision and $\mathcal{F}$-MTL represents the model trained with both sarcasm and sentiment supervisions. For fair comparisons, the network backbone of $\mathcal{F}$-MTL is the same as $\mathcal{F}$-SEN.**

| Benchmark | Method | Entire testing set | | | Subset 1 | | | Subset 2 | | |
|---|---|---|---|---|---|---|---|---|---|---|
| | | Pre(%) | Rec(%) | F1(%) | Pre(%) | Rec(%) | F1(%) | Pre(%) | Rec(%) | F1(%) |
| MUStARD | $\mathcal{F}$-SEN | 51.28 | 55.62 | 54.12 | 56.47 | 60.05 | 58.13 | 46.15 | 55.88 | 50.07 |
| | $\mathcal{F}$-MTL [4] | 52.13 | 57.42 | 54.86 | 58.24 | 62.17 | 60.43 | 42.11 | 53.95 | 46.94 |
| | △ | 0.85↑ | 1.76↑ | 0.74↑ | 1.77↑ | 2.12↑ | 2.30↑ | -4.04↓ | -1.93↓ | -3.13↓ |
| MUStARD++ | $\mathcal{F}$-SEN | 18.61 | 20.67 | 19.49 | 16.94 | 14.63 | 15.08 | 27.48 | 33.93 | 31.24 |
| | $\mathcal{F}$-MTL [4] | 19.29 | 21.33 | 19.82 | 17.61 | 19.06 | 18.41 | 26.26 | 29.77 | 27.58 |
| | △ | 0.68↑ | 0.66↑ | 0.33↑ | 0.67↑ | 4.43↑ | 3.33↑ | -1.22↓ | -4.16↓ | -3.66↓ |

divide the dataset into 16,326 training samples, 1,871 validation samples and 4,659 testing samples. As in the above setting, the sentiment label of each sample ranges from -3 to 3. The performance metrics are the same to the ones used in the CMU-MOSI dataset.

## 5.2 Implementation Details

Following Hasan et al. [11], we use different learning rates for language, acoustic and visual encoders, as well as other blocks in our approach. Specifically, the learning rate for both acoustic and visual encoders is set to 3e-3, while the learning rate for the language encoder and other blocks is set to 1e-5. Adam is adopted as the optimizer. During the training process, the learning rate decreases according to the cosine decay policy. The batch size is set to 64 in the training process. The dropout rates is set to 0.4. The training epoch is set to 50. We use early-stopping with patience of 10 to avoid overfitting. Following Hasan et al. [11], we utilize a 12-layer ALBERT, an 8-layer transformer and a 1-layer transformer to respectively process the text, visual and acoustic modalities within sarcasm/sentiment feature encoders. The number of MTF layer within the sarcasm feature encoder is set to 4. The PS2RI module utilizes 4 iterations, i.e., $J = 4$. The hidden-layer dimension is set to 192, i.e., $d^H = 192$. The models are trained on a 3090 GPU. The hyper-parameters are determined on the validation set. In experiments, we report results averaged on 5 runs with different random seeds.

## 5.3 Result

In this paper, we design four sets of experiments to answer four research questions, through which we progressively study the promoting relationship between sarcasm detection and sentiment analysis:

- **RQ1**: Can sarcasm information contribute to sentiment analysis by directly utilizing the MTL framework?
- **RQ2**: Does the proposed framework achieve performance superiority overall and on specific data splits, compared to existing baselines?
- **RQ3**: Can the proposed SASL mechanism bring performance improvement when applied to existing video sentiment recognition models?
- **RQ4**: Can the proposed SASL mechanism achieve good robustness on wider benchmarks where the sarcasm labels are not provided?

Next, we detail the answer for each question and discuss the experimental results.

**Answer to RQ1.** For RQ1, we resort to a single model (i.e., the fusion function $\mathcal{F}$ consisting of 4 MTF layers) introduced in Section 3.3 as the base model, which processes the multimodal input by incorporating modality-wise interactions. We use $\mathcal{F}$-SEN to denote the model that is only trained using the sentiment supervision. To examine whether sarcasm information contributes to sentiment analysis by directly utilizing the MTL framework, we implement $\mathcal{F}$-MTL which includes two classification heads on top of $\mathcal{F}$-SEN for generating sarcasm and sentiment predictions and is trained with both sarcasm and sentiment supervisions. For in-depth analysis, we partition the testing set into two separate sub-sets: one containing sarcastic samples (Subset 1) and the other containing non-sarcastic samples (Subset 2). Table 1 shows the performance of $\mathcal{F}$-SEN and $\mathcal{F}$-MTL on different data splits.

From Table 1, we can see that $\mathcal{F}$-MTL can only achieve a marginal improvement over $\mathcal{F}$-SEN (i.e., 54.86% *vs.* 54.12% in terms of F1 score). Moreover, this improvement is solely achieved from Subset 1 containing sarcastic samples. When it comes to Subset 2 containing non-sarcastic samples, $\mathcal{F}$-MTL is significantly inferior than the $\mathcal{F}$-SEN base model. This observation shows that **the MTL framework only improves sentiment predictions over sarcastic samples, while causing negative interference for non-sarcastic samples which can be easily identified by a single sentiment recognition model**. Hence, we speculate that the MTL framework which requires representation sharing among tasks may bring noisy information negatively affecting the sentiment prediction task. This calls for a more effective strategy to integrate sarcasm information into the sentiment prediction task.

**Answer to RQ2.** For RQ2, we compare our framework PS2RI with existing state-of-the-art video sentiment analysis baselines, including TFN [57], LMF [27], MulT [44], MISA [12], MAG [34], $\mathcal{F}$-MTL [4], GSLM+ [3] and DMD [18]. For fair comparisons, we use ALBERT as the textual encoder for all the baselines and report the performance of PS2RI and the compared baselines averaged over 5 independent runs with different random seeds in Table 2. As shown in Table 2, PS2RI outperforms all the baseline models with a large performance gain. Specifically, we can see that PS2RI can obtain consistent performance improvement over both sarcasm samples (Subset 1) and non-sarcasm samples (Subset 2). This observation shows that **PS2RI has indeed found an effective way to use sarcasm information to enhance sentiment task, which eliminates the negative impact of the MTL framework**.

**Table 2: Performance comparison on the MUStARD and MUStARD++ benchmarks. All the baselines are reproduced by using the codes provided in their papers. For fair comparisons, all the baselines utilize ALBERT as the text encoder.**

| Benchmark | Method | Entire testing set | | | Subset 1 | | | Subset 2 | | |
|---|---|---|---|---|---|---|---|---|---|---|
| | | Pre(%) | Rec(%) | F1(%) | Pre(%) | Rec(%) | F1(%) | Pre(%) | Rec(%) | F1(%) |
| MUStARD | TFN [57] | 49.36 | 45.57 | 48.39 | 53.92 | 49.41 | 50.99 | 45.63 | 47.21 | 46.44 |
| | LMF [27] | 52.14 | 54.41 | 52.66 | 58.45 | 60.78 | 58.57 | 45.46 | 48.04 | 45.58 |
| | MulT [44] | 51.89 | 55.15 | 52.83 | 54.82 | 58.82 | 56.74 | 44.41 | 50.56 | 48.90 |
| | MISA [12] | 54.07 | 49.82 | 52.93 | 60.07 | 55.62 | 58.96 | 43.57 | 48.41 | 46.13 |
| | MAG [34] | 52.76 | 54.27 | 53.85 | 62.53 | 59.74 | 59.16 | 42.56 | 50.42 | 47.32 |
| | $\mathcal{F}$-MTL [4] | 52.13 | 57.42 | 54.86 | 58.24 | 62.17 | 60.43 | 42.11 | 53.95 | 46.94 |
| | GSLM+ [3] | 52.16 | 58.12 | 55.43 | 60.21 | 62.13 | 61.47 | 45.13 | 50.49 | 48.36 |
| | DMD [18] | 54.29 | 59.08 | 56.27 | 60.71 | 62.49 | 61.71 | 44.97 | 52.82 | 49.66 |
| | PS2RI (ours) | **57.46** | **62.06** | **58.45** | **61.73** | **66.47** | **63.52** | **52.42** | **57.65** | **53.50** |
| MUStARD++ | TFN [57] | 14.62 | 20.51 | 16.95 | 9.05 | 14.90 | 12.71 | 27.33 | 30.71 | 29.35 |
| | LMF [27] | 15.24 | 19.17 | 17.76 | 13.34 | 15.63 | 14.21 | 26.86 | 23.21 | 24.86 |
| | MulT [44] | 15.96 | 25.01 | 18.03 | 14.15 | 12.37 | 13.82 | 31.34 | 25.74 | 29.21 |
| | MISA [12] | 16.01 | 25.83 | 18.56 | 15.63 | 17.19 | 15.58 | 26.67 | 30.17 | 28.62 |
| | MAG [34] | 17.46 | 22.51 | 19.38 | 17.07 | 20.31 | 18.35 | 26.22 | 35.71 | 29.76 |
| | $\mathcal{F}$-MTL [4] | 19.29 | 21.33 | 19.82 | 17.61 | 19.06 | 18.41 | 26.26 | 29.77 | 27.58 |
| | GSLM+ [3] | 19.61 | 24.17 | 20.96 | 20.13 | 17.19 | 18.44 | 32.43 | 31.14 | 31.65 |
| | DMD [18] | 20.37 | 24.16 | 22.81 | 22.14 | 17.31 | 19.04 | 34.56 | 33.46 | 34.07 |
| | PS2RI (ours) | **24.06** | **25.83** | **24.28** | **22.96** | **18.75** | **19.27** | **37.59** | **33.93** | **35.04** |

**Table 3: Results of applying the proposed SASL mechanism to existing video sentiment recognition models.**

| Benchmark | Method | Pre(%) | Rec(%) | F1(%) |
|---|---|---|---|---|
| MUStARD | MulT [44] | 51.89 | 55.15 | 52.83 |
| | with SASL | **52.38** | **58.71** | **54.41** |
| | DMD [18] | 54.29 | 59.08 | 56.27 |
| | with SASL | **57.34** | **60.57** | **58.03** |
| MUStARD++ | MulT [44] | 15.96 | 19.17 | 17.76 |
| | with SASL | **17.03** | **20.11** | **19.34** |
| | DMD [18] | 20.37 | 24.16 | 22.81 |
| | with SASL | **21.12** | **25.34** | **23.34** |

**Table 4: Result of applying the proposed SASL mechanism to existing video sentiment recognition models on the CMU-MOSI and CMU-MOSEI benchmarks.**

| Benchmark | Method | ACC$_7$(%) | ACC$_2$(%) | F1(%) |
|---|---|---|---|---|
| CMU-MOSI | MulT [44] | 39.44 | 82.19 | 81.94 |
| | with SASL | **41.25** | **83.79** | **83.64** |
| | DMD [18] | 43.88 | 83.77 | 84.35 |
| | with SASL | **45.36** | **86.17** | **85.94** |
| CMU-MOSEI | MulT [44] | 51.23 | 81.41 | 81.26 |
| | with SASL | **52.91** | **82.75** | **82.93** |
| | DMD [18] | 52.29 | 84.51 | 84.57 |
| | with SASL | **54.45** | **86.82** | **86.72** |

**Answer to RQ3.** For RQ3, we conduct experiments by applying the proposed Sarcasm-Aware Sentiment Learning (SASL) mechanism to other existing video sentiment recognition models (including the typical MulT [44] baseline and the state-of-the-art DMD [18] baseline). Specifically, the SOSR layer and SI layer are added before the each layer of the MulT model and DMD model. Table 3 shows the corresponding results averaged over 5 different runs. We can see that **our proposed SASL mechanism also achieves consistent performance improvement based on other baselines, which demonstrates the scalability of the SASL mechanism.**

**Answer to RQ4.** For RQ4, we conduct experiments by applying the SASL mechanism to the MulT [44] and DMD [18] models on CMU-MOSI and CMU-MOSEI, which are two extensively used benchmarks in video sentiment analysis. As these two datasets do not contain sarcasm labels, we directly adopt the sarcasm feature

**Table 5: Ablation study for the contribution of each design on the MUStARD and MUStARD++ benchmarks. The results are reported in terms of the weighted-F1.**

| Benchmark | Method | Entire testing set | Subset 1 | Subset 2 |
|---|---|---|---|---|
| MUStARD | PS2RI (full model) | **58.45** | **63.52** | **53.50** |
| | w/o SI | 54.12 | 62.89 | 47.58 |
| | w/o SOSR | 54.60 | 58.63 | 52.17 |
| MUStARD++ | PS2RI (full model) | **24.28** | **19.27** | **35.04** |
| | w/o SI | 20.07 | 18.11 | 24.75 |
| | w/o SOSR | 20.94 | 17.35 | 26.33 |

extraction module trained on MUStARD++ to generate sarcasm features for their samples. Table 4 shows the corresponding results averaged over 5 different runs. We can see that our proposed SASL mechanism can also lead to consistent performance improvements on both CMU-MOSI and CMU-MOSEI benchmarks. **This observation clearly shows the robustness of our approach on settings where the sarcasm labels are not provided.**

## 5.4 Analysis

**Ablation Study.** We report the ablation study results on the MUStARD and MUStARD++ benchmarks in Table 5. The first row of each dataset displays the performance of the full model. In the second row of each dataset, we remove the SI blocks from the full model, resulting in the $\mathcal{F}$-SEN base model which only involves the sentiment information. The consistent performance drops observed on both MUStARD and MUStARD++ show that the PS2RI model can effectively utilize the sarcasm information to improve the sentiment recognition task. Compared with the MTL framework, incorporating the sarcasm information within the PS2RI framework does not adversely affect the sentiment prediction over non-sarcastic samples. In the third row of each dataset, we remove the SOSR blocks from the full model by directly integrating the original sarcasm feature generated from the Sarcasm Feature Extraction Module into sentiment features. The performance degradation compared to that of the first row can indicate that the original sarcasm feature can bring negative interference for sentiment analysis. This observation clearly demonstrates the effectiveness of modeling

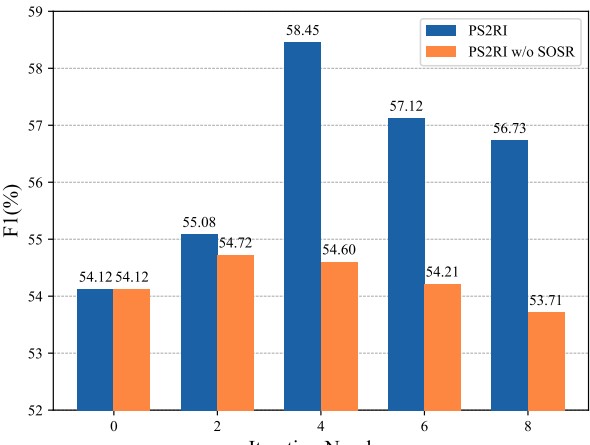

**Figure 4: Performance under different iteration numbers within the SASL module.**

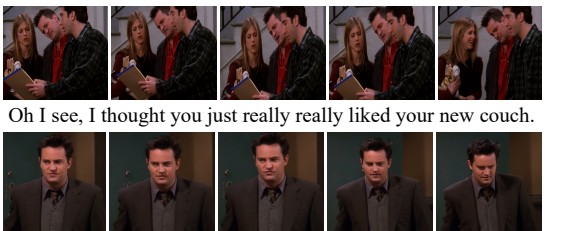

Oh I see, I thought you just really really liked your new couch.

No  thanks Josephine!

**Figure 5: Case samples with sarcasm intention, for which the $\mathcal{F}$-SEN base model fails to recognize the true sentiments.**

sentiment-oriented sarcasm features via SOSR blocks, which is consistent with our motivation.

**Analysis on Sarcasm Features.** In order to further investigate the sentiment-oriented sarcasm refinement mechanism of the proposed approach, we respectively train two sentiment classification heads on vanilla sarcasm features $Z_{sar}$ extracted from the SFE module and sentiment-oriented sarcasm features $\hat{Z}_{sar}$ obtained from the last layer of SOSR. The performance on $\hat{Z}_{sar}$ is significantly worse than that on $Z_{sar}$ (i.e., 46.15% vs 50.42% in terms of weighted-F1 on MUStARD), which validates that our approach can effectively discard information irrelevant to the sentiment analysis task from sarcasm features, resulting in sentiment-oriented sarcasm features to enhance the sentiment task.

**Iteration Numbers within SASL.** Furthermore, we test the performance by varying the iteration number within SASL. For each number, we also conduct experiments with SOSR blocks removed (i.e., directly integrate the original sarcasm feature extracted from the SFE module into sentiment features). The corresponding results are shown in Figure 4. In general, the proposed PS2RI framework can achieve gradually increasing performance when more interactive iterations are utilized to learn sarcasm-aware sentiment feature. However, for PS2RI w/o SOSR, the performance improvement is limited when more than 2 SI blocks are utilized. This observation further demonstrates that modeling sentiment-oriented sarcasm features is important for effectively improving sentiment recognition. Moreover, a performance drop is observed when the iteration

**Table 6: Results of applying the SASL mechanism to existing video sentiment recognition models. We keep an equal number of parameters in models *with more layers* and *with SASL***

| Benchmark | Method | Pre(%) | Rec(%) | F1(%) |
|---|---|---|---|---|
| MUStARD | MulT [44] | 51.89 | 55.15 | 52.83 |
| | with more layers | 51.43 | 54.34 | 52.17 |
| | with SASL | **52.38** | **58.71** | **54.41** |
| | DMD [18] | 54.29 | 59.08 | 56.27 |
| | with more layers | 52.17 | 57.17 | 55.06 |
| | with SASL | **57.34** | **60.57** | **58.03** |
| MUStARD++ | MulT [44] | 15.96 | 19.17 | 17.76 |
| | with more layers | 15.01 | 18.21 | 17.33 |
| | with SASL | **17.03** | **20.11** | **19.34** |
| | DMD [18] | 20.37 | 24.16 | 22.81 |
| | with more layers | 20.12 | 22.13 | 21.23 |
| | with SASL | **21.12** | **25.34** | **23.34** |

number is greater 4. This observation indicates that the sentiment-oriented sarcasm information can be well modeled by using 4 iteration numbers and more network parameters will lead to overfitting.

**Case Study.** To illustrate the assistance of sarcasm information, we provide case studies which express sarcastic intention in Figure 5. Take the top sample of Figure 5 as example, the word "like" in the text and the smiling face in the last video clip convey the positive sentiment, while the confused face in the first four video clips conveys the negative sentiment. The conflict sentiments constitute the sarcastic expression. Without the assistance of sarcasm information, the $\mathcal{F}$-SEN base model fails to recognize their sentiment polarities. On the other hand, by explicitly modeling sarcasm-aware sentiment features, the PS2RI approach can effectively recognize the true sentiment.

**Comparison with Equal Parameter Number.** In the Results section of the main paper, we implement the proposed SASL mechanism on the MulT and DMD models and achieve a clear performance improvement. To elucidate the attained performance improvement more distinctly, we augment the original MulT and DMD models with external layers, maintaining an equal number of model parameters with the SASL variants. From Table 6, we can see that the models with additional layers exhibit inferior performance compared to the original ones. This observation emphasizes that the performance improvement of our approach cannot be solely ascribed to the additional parameters introduced by the SASL mechanism.

## 6 Conclusion & Future work

In this study, we investigate how to enhance sentiment analysis through the assistance of sarcasm detection. To this end, we discuss the limitations associated with the current MTL-based approaches and propose the PS2RI model to iteratively merge sarcasm-related features into the sentiment learning process. In order to alleviate the negative interference of sarcasm detection, we introduce Sentiment-Oriented Sarcasm Refinement blocks which model sentiment-oriented sarcasm features by attending sarcasm features to sentiment features and Sarcasm Integration blocks which incorporate sentiment-oriented sarcasm features into sentiment features. By applying the SASL mechanism to CMU-MOSI and CMU-MOSEI datasets, we demonstrate the robustness of our approach in settings where sarcasm labels are absent. However, there still exist domain gaps between the source and target datasets. In future work, we will address the challenge by integrating domain adaptation methods into the training phase.

## Acknowledgments

This project is supported by the National Natural Science Foundation of China Grant (No. 62106204), the Frontier Cross Innovation Team Project of Southwest Jiaotong University under Grant (No. YH1500112432297), the Natural Science Foundation of Sichuan under Grant (No. 2022NSFSC0911), the Fundamental Research Funds for the Central Universities (NO. 2682024ZTPY055), and the Engineering Research Center of Sustainable Urban Intelligent Transportation, Ministry of Education.

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
