# OpenReview forum: "Sentiment-Oriented Sarcasm Integration: Effective Enhancement of Video Sentiment Analysis with Sarcasm Assistance"
_acmmm.org/ACMMM/2024/Conference — MM2024 Poster_

### Official Review · Reviewer_UNRj · 2024-05-06

**Rating:** 2
**Confidence:** 3

**Summary:**

The paper is about the incorporation of sarcasm information to enhance video sentiment analysis. It proposes the Progressively Sentiment-oriented Sarcasm Refinement and Integration (PS2RI) framework, which focuses on modeling sentiment-oriented sarcasm features to improve sentiment prediction in videos. The framework iteratively performs the sentiment-oriented sarcasm refinement and sarcasm integration operations within the sentiment recognition framework, progressively learning sarcasm-aware sentiment features without suffering from detrimental interplays caused by irrelevant information.

**Strengths:**

The framework of this model is easy to understand, and its effectiveness has been verified through extensive experiments in the paper.

The proposed model in the paper achieved the best results on four datasets.

**Limitations:**

The training process for the two encoders is unclear.

The extensive use of methods from paper [1] for feature preprocessing and encoding raises some concerns, as I have noticed that paper [1] originates from a different multimodal domain. Applying these methods directly to the field of video sentiment analysis may not necessarily yield the optimal results.

I noticed that as the hyperparameter Iteration Number increases, the performance does not seem to reach a ceiling. Can it be further improved?

There are misspelled words and errors in the mathematical notation or symbols in the paper, such as in Step 5 of Algorithm 1.

Please consider providing access to the code utilized in the paper to facilitate validation and reproducibility.

I will consider revising the review scores.

[1] Humor Knowledge Enriched Transformer for Understanding Multimodal Humor. AAAI.

**Suitability:**

3

---

### Official Review · Reviewer_hSDv · 2024-05-14

**Rating:** 4
**Confidence:** 3

**Summary:**

The research paper delves into enhancing video sentiment analysis by integrating sarcasm information, a commonly overlooked aspect in current studies. It introduces the Progressively Sentiment-oriented Sarcasm Refinement and Integration (PS2RI) framework to capture sentiment-oriented sarcasm features for enhanced sentiment prediction. The study underscores the challenges of Multi-Task Learning (MTL) methods in managing sarcasm detection and sentiment analysis simultaneously. Through iterative refinement of sarcasm features with sentiment information and their integration into sentiment learning, PS2RI effectively mitigates the negative impact of sarcasm detection on sentiment analysis. Extensive experimental results confirm the efficacy and scalability of the proposed method, underscoring the significance of incorporating sarcasm into video sentiment analysis for more accurate predictions.

**Strengths:**

Following are the strengths of the work:
1. Innovative Framework: The paper introduces the novel Progressively Sentiment-oriented Sarcasm Refinement and Integration (PS2RI) framework, which effectively addresses the challenges of incorporating sarcasm information into video sentiment analysis.

2. Iterative Approach: The iterative nature of the PS2RI framework, where sarcasm features are refined using sentiment information and integrated into sentiment learning, is a notable strength. This iterative process allows for the progressive learning of sarcasm-aware sentiment features, ultimately improving sentiment prediction accuracy.

3. Empirical Validation: The study conducts extensive experiments to validate the effectiveness and scalability of the proposed PS2RI framework. The paper demonstrates the superiority of the new approach over conventional MTL methods, emphasizing the importance of considering sarcasm in video sentiment analysis for more accurate predictions.

4. Clear Communication: The clarity in presenting the methodology and findings enhances the paper's readability and understanding for readers interested in sentiment analysis and sarcasm detection in videos.

**Limitations:**

Following are the points where the research work requires further discussion:
Theoretical Foundation: While the paper introduces a novel framework for integrating sarcasm information into sentiment analysis, it would benefit from a deeper exploration of the underlying principles governing sarcasm detection and sentiment analysis. Strengthening the theoretical background through illustrative examples would bolster the author's motivation and rationale.

Generalizability: The paper could expand its discussion on the generalizability of the proposed PS2RI framework across different datasets or domains. Addressing potential limitations or challenges in applying the framework to various scenarios would enhance the paper's practical relevance and usability.

Methodology: How were features extracted for the three modalities? The explanation of feature preprocessing lacks sufficient justification. Moreover, why was a particular research paper [Hasan et al., AAAI, 2021] chosen as the foundation for this work? This aspect requires a more thorough discussion.

Implementation Details: There are a few missing citations, and the rationale behind adopting specific implementation settings from a previous research paper [Hasan et al., AAAI, 2021] needs clarification.

Qualitative Analysis: Although the proposed framework demonstrates its superiority quantitatively, the qualitative analysis is lacking depth, especially regarding the case study (Figure 5), which does not effectively illustrate the superiority of PS2RI. Furthermore, the discussion on error analysis of the current model is absent.

Real-world Applications: The paper would benefit from a discussion on the real-world applications and implications of the PS2RI framework. Exploring potential enhancements to the framework in future research could offer valuable insights for researchers in the same field.

**Suitability:**

3

---

### Official Review · Reviewer_iPtN · 2024-05-20

**Rating:** 5
**Confidence:** 3

**Summary:**

This paper aims to address the issue of overlooking sarcasm in video sentiment analysis. The proposed approach, named Progressively Sentiment-oriented Sarcasm Refinement and Integration (PS2RI), focuses on modeling sentiment-oriented sarcasm features to enhance sentiment prediction. PS2RI iteratively performs sentiment-oriented sarcasm refinement and sarcasm integration within the sentiment recognition framework, aiming to progressively learn sarcasm-aware sentiment features without suffering from detrimental interactions caused by irrelevant information. Extensive experiments are conducted to validate the effectiveness and scalability of the proposed PS2RI framework.

**Strengths:**

1.	PS2RI represents a novel approach to incorporate sarcasm information into video sentiment analysis. The iterative refinement and integration of sarcasm features within the sentiment recognition framework is a creative solution to the issue of detrimental interactions between sarcasm detection and sentiment analysis.
2.	PS2RI covers both sentiment-oriented sarcasm refinement and sarcasm integration, demonstrating a comprehensive approach to addressing the challenges of sarcasm in sentiment analysis. The iterative nature of the framework allows for progressive learning of sarcasm-aware sentiment features.
3.	The paper is comprehensive, providing ample discussion and analysis. The Methodology and Experiment sections are detailed, and the language, figures, and tables throughout the paper are logically clear.

**Limitations:**

1.	Is there a distinction between the Sarcasm-Aware Sentiment Module (SASM) and Sarcasm-Aware Sentiment Learning (SASL) in the paper? The author is expected to provide further clarification on this matter.
2.	The paper lacks a table outlining the sample label division of the datasets. The author is encouraged to include this information for completeness.
3.	The Case Study is described in a rather concise manner. It is requested that the author analyze the characteristics of the cases presented in the case study and explain why the proposed model is effective in identifying genuine sentiment.
4.	I am unclear about the explanation in the Answer to RQ4. The CMU-MOSI and CMU-MOSEI datasets do not have sarcasm labels, yet the SASL mechanism is based on sarcasm detection. Even though the sarcasm feature extraction module trained on MUStARD++ is applied to these datasets, the final labels are not sarcasm versus non-sarcasm. What is the significance of robustness in this context? The author is requested to provide a detailed and supplementary explanation on this matter.

**Suitability:**

3

---

### Official Review · Reviewer_eLuM · 2024-05-24

**Rating:** 2
**Confidence:** 4

**Summary:**

The study recognizes the significance of sarcasm in video sentiment analysis, which is often overlooked. This focus provides a unique perspective and fills a gap in the current research.The proposed PS2RI framework iteratively performs sentiment-oriented sarcasm refinement and integration, which allows for a progressive learning of sarcasm-aware sentiment features. This approach avoids the detrimental interplay between sarcasm detection and sentiment analysis tasks that can occur in traditional multi-task learning frameworks.

**Strengths:**

1. Extensive experiments are conducted to validate both the effectiveness and scalability of the PS2RI framework. This ensures that the results are robust and applicable to a wide range of scenarios.
2. The authors give detialed presentation of feature process and experiement settings.

**Limitations:**

1. The title of this paper is strange and the authors may reconsider the proper title that can represent the overall innovation.
2. Sarcasm is a complex linguistic phenomenon that can be difficult to detect, especially in multimodal contexts such as videos. The study's performance in sarcasm detection and refinement may therefore be limited.
3. While the study demonstrates the effectiveness of the PS2RI framework, it does not provide deep insights into why and how it works. Additional qualitative and interpretative analysis could help to further strengthen the approach.

**Suitability:**

3

---

### Meta-Review · Area_Chair_Cgjz · 2024-07-02

**Recommendation:** Accept (Poster)
**Confidence:** 5

**Metareview:**

This paper proposes a novel framework called PS2RI for incorporating sarcasm information into video sentiment analysis. Reviewers acknowledge the innovative approach and extensive experiments. Some questions remain such as the lack of deep insights into the framework's effectiveness, unclear distinctions between key components, insufficient qualitative analysis, and questions about the generalizability of the approach. Two reviewers increased their scores after the rebuttal, where the overall assessment lean towards accept. Therefore, despite the concerns, the topic studied in this paper has good potential, and accept is recommended. The authors are encouraged to address the reviewers' concerns thoroughly in the camera-ready version.